# Discovery of Selective SIRT2 Inhibitors as Therapeutic Agents in B-Cell Lymphoma and Other Malignancies

**DOI:** 10.3390/molecules25030455

**Published:** 2020-01-21

**Authors:** Sarwat Chowdhury, Smitha Sripathy, Alyssa A. Webster, Angela Park, Uyen Lao, Joanne H. Hsu, Taylor Loe, Antonio Bedalov, Julian A. Simon

**Affiliations:** 1Clinical Research Division, Fred Hutchinson Cancer Research Center, Seattle, WA 98109, USA; csarwat@yahoo.com (S.C.); ssripath@fredhutch.org (S.S.); awebster@fredhutch.org (A.A.W.); aypark12@gmail.com (A.P.); ulao@fredhutch.org (U.L.); jhhsu11@gmail.com (J.H.H.); taylor.k.loe@gmail.com (T.L.); abedalov@fredhutch.org (A.B.); 2Human Biology Division, Fred Hutchinson Cancer Research Center, Seattle, WA 98109, USA

**Keywords:** sirtuin, SIRT2, acetylation, cancer, lymphoma

## Abstract

Genetic ablation as well as pharmacological inhibition of sirtuin 2 (SIRT2), an NAD^+^-dependent protein deacylase, have therapeutic effects in various cancers and neurodegenerative diseases. Previously, we described the discovery of a dual SIRT1/SIRT2 inhibitor called cambinol (IC_50_ 56 and 59 µM, respectively), which showed cytotoxic activity against cancer cells in vitro and a marked anti-proliferative effect in a Burkitt lymphoma mouse xenograft model. A number of recent studies have shown a protective effect of SIRT1 and SIRT3 in neurodegenerative and metabolic diseases as well as in certain cancers prompting us to initiate a medicinal chemistry effort to develop cambinol-based SIRT2-specific inhibitors devoid of SIRT1 or SIRT3 modulating activity. Here we describe potent cambinol-based SIRT2 inhibitors, several of which show potency of ~600 nM with >300 to >800-fold selectivity over SIRT1 and 3, respectively. In vitro, these inhibitors are found to be toxic to lymphoma and epithelial cancer cell lines. In particular, compounds **55** (IC_50_ SIRT2 0.25 µM and <25% inhibition at 50 µM against SIRT1 and SIRT3) and **56** (IC_50_ SIRT2 0.78 µM and <25% inhibition at 50 µM against SIRT1 and SIRT3) showed apoptotic as well as strong anti-proliferative properties against B-cell lymphoma cells.

## 1. Introduction

Sirtuins are NAD^+^-dependent protein deacylases (this term is more accurate than the previously widely used “deacetylases” since these enzymes remove other acyl modifications) that have emerged as key regulators of diverse cellular processes such as chromatin modification, gene expression, DNA repair, cell cycle control and cell survival [1,2]. The human genome encodes seven isoforms, sirtuins 1–7 of which sirtuin 2 (SIRT2) is primarily a cytoplasmic protein that is known to localize to the nucleus especially during mitosis [3]. SIRT2 exhibits robust deacetylase and demyristoylase activities against histone and non-histone substrates. Expressed ubiquitously, SIRT2 is prominently expressed in brain and muscle tissues [4]. Genetic ablation of SIRT2 has been shown to have protective effects in models of neurodegenerative disorders such as Huntington’s [5] and Parkinson’s diseases [6]. A growing body of literature also suggests that SIRT2 may be a valid drug target in certain cancers. SIRT2-mediated deacetylation has been shown to promote mutant K-ras oncogenic activity [7]. Similarly, SIRT2 enhances the stability of n-myc and c-myc oncogenes leading to cancer cell proliferation [8]. Further, siRNA-mediated downregulation of SIRT2 was found to induce apoptosis in gliomas [9], and hepatocellular and pancreatic carcinomas [10].

SIRT2 is known to process diverse substrates such as histone H4 lysine-16 (H4K16) [11], alpha-tubulin [12], p53 [13], EP300 [14]. Foxo-family of proteins [14,15] and glucose-6-phosphate dehydrogenase (G6PD) [16] among others. Owing to its regulatory roles in various cellular processes, perhaps it is not surprising that the dual roles of SIRT2 as a cancer suppressor as well as an oncogene [17] have been reported [18]. Recent consensus seems to reconcile these contradictory findings by taking into account disease-specific contexts such as cancer sub-type, expression pattern of SIRT2 and its substrate proteins as well as their roles in oncogenic signaling [19,20]. As with SIRT2, there are conflicting reports of cancer suppressor and oncogenic activities of SIRT1 and 3 [21] in various tumor types [22,23]. Additionally, SIRT1 activity has been demonstrated to confer neuroprotection in several age-related neurodegenerative disorders including Parkinson’s and Alzheimer’s diseases [24,25] and metabolism related disorders [26]. The mitochondrial isoform SIRT3 deacylase activity is important for mitochondrial metabolism, and cell survival [27]. Additionally, loss of function studies indicated SIRT3 has a tumor suppressor function through modulation of reactive oxygen species (ROS) production [28]. Because of these potentially confounding activities, identification of isoform-selective sirtuin inhibitors is the most practical approach to validating SIRT2 as a drug target.

Historically, it has been difficult to synthesize isoform-selective sirtuin inhibitors primarily because all sirtuin isoforms possess a core 260-amino acid catalytic domain which is highly conserved from bacteria to humans [29,30]. The SIRT1 catalytic domain is 45% identical to SIRT2 and bears 69% similarity [31]. Previous reported sirtuin inhibitors showed little or no isoform selectivity and had modest potency. Nonetheless, dual SIRT1/2 inhibitors showed anti-cancer activity. For example, EX527, sirtinol, and saleramide induced cell death in breast cancer cell line in vitro in a p53 dependent manner [32]. Splitomicin [33] and its analogs showed anti-proliferative properties in MCF-7 breast cancer cell line [34]. A thio-myristoylated dipeptide inhibitor of SIRT2 with nanomolar potency was found to promote degradation of c-myc and have anti-cancer activity in multiple cell lines [35]. In neurodegeneration models, AGK2, a sirtuin inhibitor with higher selectivity for SIRT2 (IC_50_ for SIRT1 >50 µM and SIRT2 23.5 µM, respectively) showed protection from alpha-synuclein toxicity in Parkinson’s disease model [6]. Encouragingly, recent reports of sub-micromolar SIRT2-selective inhibitors also show neuroprotection in a Parkinson’s disease model [36,37,38].

We previously reported a dual SIRT1/2 inhibitor called cambinol **1** (in vitro IC_50_ 54 and 46 µM respectively) [39] that sensitized cells to paclitaxel and etoposide and, was cytotoxic as a single agent against B-cell lymphoma cell lines in vitro and xenograft models in vivo. As expected from the distinct mechanism and no sequence homology between class I and II deacetylases, cambinol had no activity against class I (HDAC1) or II (HDAC6) enzymes. Intriguingly, a follow-up study with cambinol-based dual SIRT1/2 inhibitors demonstrated a strong correlation between SIRT2 inhibitory activity and cytotoxicity in Namalwa Burkitt lymphoma cell line [40]. Following our long-standing interest in anti-lymphoma agents, coupled with the need for isoform-selective sirtuin modulators, we launched a medicinal chemistry effort to identify SIRT2 selective cambinol analogs and evaluate cytotoxicity against B-cell lymphomas. Herein we report the discovery of open ring cambinol analogs, devoid of thio-pyrimidinone moiety as SIRT2-specific inhibitors (Figure 1). Based on structure-activity relationship (SAR) evaluation, potency was improved >200 fold; displaying IC_50_ at the sub-micromolar level as racemates. Concurrently, high selectivity against SIRT1 and 3 isoforms was also achieved. These compounds were found to be cytotoxic against B-cell lymphoma and epithelial cancer cell lines in vitro.

## 2. Results

Our previous efforts to synthesize and evaluate tetracyclic cambinol analogs as sirtuin inhibitors showed that replacement of thio-pyrimidinone ring with other heterocycles (Figure 1) can impart improved potency and isoform-selectivity, as exemplified by pyrazolone compound **2**.

While exploring variants of the heterocyclic ring of cambinol keeping the hydroxynaphthalene and phenyl ring moieties intact, we discovered that simple β-ketoamide **4**, generated by aminolysis of lactone **3** showed improved inhibitory potency (IC_50_ SIRT2 13 µM) as well as high selectivity (IC_50_ SIRT1, >200 µM) (Scheme 1). We hypothesized that by the virtue of open chain scaffold, these chiral cambinol analogs might display enhanced receptor/small molecule interaction as compared to **1**. Realizing the potential for rapid diversification of amide sidechains as a means to improve potency, selectivity and to modulate physicochemical properties of these compounds, we embarked on an SAR study to identify potent and isoform-selective SIRT2 inhibitors. On account of keto-enol tautomerization leading to racemization, we commenced an investigation with racemates of the desired cambinol analogs.

### 2.1. Chemistry

Scheme 2 shows a general synthesis route to 2-hydroxy-1-naphthaldehydes **7**. Our general synthetic scheme calls for various substituted hydroxynaphthaldehydes **7** and β-keto esters (Scheme 3) as the starting materials.

Titanium-catalyzed Rieche formylation of commercially available substituted naphthols was used to generate the corresponding naphthaldehydes **7** in good yields (see Appendix A). Hydroxyquinoline carbaldehyde **8** was obtained in two steps first by Reimer-Tiemann reaction 2,6-dihyoxyquinoline followed by chlorination with POCl_3_ to give the desired product in quantitative yield over two steps.

While most β-keto esters were obtained from commercial sources, compound **11** was prepared in two steps from nitrobenzoyl propionate ester **10**. Iron-mediated reduction of the nitro group first gave the corresponding aniline in good yield. In the subsequent step, acetylation of the aniline furnished the desired compound **11**(see Appendix A).

The core scaffolds were synthesized by one-pot morpholine-catalyzed Knoevenagel condensation/lactonization of hydroxynaphthaldehydes **12** with β-keto esters. The resultant coumarin lactones **13**, which precipitated from the reaction mixture, were obtained in good yields (see Appendix A). Selective 1,4-reduction by treatment with NaBH_4_ in anhydrous pyridine furnished the desired saturated lactones, which were subjected to aminolysis to yield the desired products **15** (Scheme 4).

### 2.2. Structure Activity Relationship Study (SAR)

The newly synthesized open chain cambinol analogs were evaluated for in vitro SIRT1, 2 and 3 inhibitory potency using commercially available purified sirtuin enzymes (Cayman Chemical, Ann Arbor, MI, USA) and luciferin-based SIRT-Glo assay (Promega Corp., Madison, WI, USA) employing a p53 acetylated peptide as substrate. In the initial screening, single point inhibition was determined using 50 µM test compound concentration to assess the potency and selectivity. We used cambinol as a positive control showing 54% SIRT1 inhibition, 46% SIRT2, and 16% SIRT3 inhibition at 50 μM (Table 1) was used. Promising compounds were evaluated by a full dose-response curve to ascertain IC_50_ against each of the sirtuin isoforms.

Our exploration began with variation of phenyl ring substituents while keeping the 6-bromo-2-hydroxy naphthyl group unchanged due to its favorable effect on isoform selectivity; a rationale we derived from our previous study. SAR of the phenyl group (Table 1) showed a clear trend where 4-substitution was favored over 3-position, and 2-substitution was found to be detrimental for inhibitory activity (e.g., **4**, **16** and **17**). Inhibitory potency against SIRT2 increased with 4-substitution of progressively larger/more hydrophobic groups (**19**, **20** and **22**) while showing no measurable activity against SIRT1 at 50 µM. For selectivity against phylogenetically closest ortholog mitochondrial SIRT3 [41] 4-CF_3_ group (e.g., **19**), imparted the highest selectivity for SIRT2 versus SIRT3 (100 fold). Only 25-fold selectivity for SIRT2 was observed both for Cl and Br-groups (**20** and **22**).

SAR of the naphthyl substituents of the open ring analogs showed that substitution at 6-position by hydrophobic group was crucial for activity (Table 2; compounds **23**, **33**, **34** vs. **29)**. Interestingly, cambinol **1**, which lacks a substituent on the naphthyl ring inhibits SIRT2 moderately (IC_50_ 59 µM).

This discrepancy could arise from a different binding mode of the acyclic compounds relative to cambinol. The hydroxyquinoline group in **35** seems to be tolerated by SIRT2. However, this analog showed lower potency than corresponding hydroxy-naphthalene compounds. Several other substitution patterns were also evaluated (**29**–**32**) and did not yield potent inhibitors of either SIRT1 or 2.

Having found that substituents 4-Br, -Cl and -CF_3_ in the phenyl ring in this open-ring scaffold give 20 to 30-fold higher potency over cambinol **1**, we sought to identify the side chains that can confer favorable SIRT2 selectivity and improve potency (Table 3). Various amines such as acyclic, carbocyclic (data not shown) and heterocyclic amines were evaluated. It was observed that bulky sidechains were detrimental for activity and conversely and sidechains derived from small acyclic amines (**33**) and those terminated in polar head groups augmented potency (e.g., **36**, **43** and **45)**. For the 3-bromophenyl series, aliphatic acyclic substituents improved not only potency, but also selectivity against SIRT1; keeping selectivity over SIRT3 unchanged (**23** vs. **36**, **43**, and **44**).

We next incorporated optimal side chain (R^2^ substituents) into the 6-bromonaphthyl and 4-substituted phenyl core scaffold (Table 3). Gratifyingly, we obtained compounds with sub-micromolar potencies against SIRT 2 while displaying high selectivity against SIRT1 and SIRT3. As seen earlier (Table 1) 4-CF_3_ substitution gave highly potent compounds **49**, **53** and **56** that showed inhibitory potency below 300 nM as a racemic mixture with >400 over SIRT1 and >800-fold selectivity over SIRT3. In general, with any given side chain, 4-CF_3_ and 4-Br groups yielded the most potent SIRT2 inhibitors as shown in Table 4.

### 2.3. Biological Evaluation of Sirtuin 2 Inhibitors

To ascertain SIRT2 inhibitory activity of our newly synthesized compounds in a cellular context, we examined acetylation level of α-tubulin, a known SIRT2 substrate [12]. Non-small cell lung cancer cell line NCI-H460 was treated with compound **55** at 0, 5, and 10 µM for 18 h. Cells were harvested, lysed and α-tubulin acetylation was analyzed by western blot (Figure 2). As predicted, dose-dependent increase in tubulin acetylation was observed upon exposure to **55** as compared to the vehicle-treated negative control confirming inhibitory activity of the newly developed SIRT2 inhibitors in cells.

With selective SIRT2 inhibitors in hand, we proceeded to evaluate cytotoxicity of our compounds in Daudi, Raji (Burkitt lymphoma) and OCI-Ly8-LAM-53 (OCI, a representative of several Diffuse Large B-Cell Lymphoma lines; DLBCL) cell lines. Briefly, cells were plated in 96-well plates and treated with inhibitors at desired concentrations for 72 h. Cellular viability was determined by measuring the amount of cellular ATP using CellTiter-Glo assay kit (Promega) and normalized against DMSO-treated controls. While all SIRT2 inhibitors were found to be toxic to B-cell lymphoma cell lines (Table 5), interestingly, primary carboxamide side chain containing inhibitors (e.g., **24**, **22**), which showed lower potencies in in vitro enzyme assays, displayed higher cytotoxicity as compared to other more potent inhibitors with polar side chains. Conversely, 4-CF_3_ phenyl substituted compounds displaying high in vitro SIRT2 inhibitory activity (e.g., **49** and **55**) were found to be less potent in the cell-based assay. *N*-methyl side chain containing compounds (**33** and **56**) appear to have more specific activity against SIRT2 and exhibit potent cellular cytotoxicity. Selected inhibitors were further evaluated in an expanded panel of cancer cell lines including breast cancer (MCF-7 and MDA-MB231), prostate cancer (PC-3) and glioblastoma multiforme (U-251). SIRT2-specific inhibitors were found to be less toxic against these cell lines compared to lymphoma cell lines. However, the trend in potency as a function of the structure is consistent with the results obtained against lymphoma cell lines. These results demonstrate toxicity of the new SIRT2 inhibitors in various cancers cell lines, B-cell lymphomas in particular, as observed with the hit compound cambinol (**1**). Interestingly, the published SIRT2 inhibitor AGK2 was only weakly active against the OCI DLBCL cell line (LC_50_ 73.2 µM).

### 2.4. SIRT 2 Inhibitor as Anti-Proliferation and Apoptosis Inducing Agent in OCI-Ly8-LAM53 (OCI) Cells

Measurement of ATP, while an excellent surrogate marker for cell viability, does not shed light on the mechanism of drug toxicity such as drug-induced cell death. To gain a better insight pertaining to the cytotoxic effects of SIRT2 inhibition, we probed for induction of apoptosis by western blot analysis as well as by annexin V staining assay upon exposure to the SIRT2 inhibitor [39]. OCI DLBCL cells were treated with **56** for 24 h. Cells were lysed and analyzed by western blot assay for cleaved poly-ADP-ribose polymerase (PARP), a well-accepted apoptosis marker [42] (Abcam, Cambridge, UK). The result (Figure 3e) showed dose-dependent PARP cleavage with progressively higher concentrations of **56** indicating SIRT2 inhibition promotes caspase-mediated apoptotic cell death. A similar finding, induction of apoptosis (Q2) was observed by annexin V staining upon exposure to **56**(Figure 3a–d) while showing negligible cell population undergoing necrotic cell death (Q1). Taken together, these results show that SIRT2 inhibition ensues dose-dependent induction of apoptosis.

## 3. Discussion

Our initial effort to identify SIRT2-selective cambinol analogs has led to the discovery of open chain inhibitors derived from cambinol displaying submicromolar potency against SIRT2 and high selectivity against SIRT1/3 in vitro. In this initial SAR, we explored substitution of three functional groups: the phenyl group (box A), naphthyl group (box B) and the side chain (box C) (as shown in Figure 4) while beta-keto and the hydroxyl group were kept constant. The beta-keto amide functionality makes these inhibitors susceptible to racemization at the chiral center under physiological conditions. Because of this, all of the compounds described herein were tested as racemates.

SAR study revealed that 4-substituted phenyl group (box A) and 6-substituted 2-hydroxynaphthyl (box B) connected by a three-carbon linker containing a ketone, and a substituted aliphatic amide functionality (box C) (Figure 4) provides a hydrophobic core of the most potent SIRT2 inhibitors. Docking studies with cambinol and its analogs published earlier concluded [43] that cambinol and analogs bind in the substrate binding narrow cleft where the hydroxyl group of the naphthyl ring system (box B) forms a key interaction with Glu116 in SIRT2.

A corollary drawn from the binding orientation [43] predicts that interaction with Gln167 and His187 in SIRT2 would impart both potency and selectivity since these residues are unavailable in the more open binding pocket of SIRT1. Our observation from the SAR with the beta-keto amide side chains (box C) agrees well the docking study showing that small polar sidechains enhance potency and impart high selectivity against highly homologous SIRT1 and SIRT3. As a result, we were able to improve selectivity for STIR2 by more than 200-fold over cambinol (**1**). Interestingly, lack of symmetry in the three dimensional SIRT2 binding pocket [43] suggests that the potency of the two enantiomers of SIRT2 inhibitors would be different. This remains to be verified with non-enolizable cambinol analogs.

Our previous reports [39,40] and studies by other groups have shown that pharmacological inhibition or genetic downregulation of SIRT2 has an anti-cancer effect in various malignancies which corroborates our observation in the present study [44,45,46]. Newly synthesized cambinol analogs having high specificity for SIRT2 were found to be toxic to in a panel of cancer cell lines. Further, as with cambinol, higher toxicity against B-cell lymphoma cell lines such as in Raji and Daudi (Burkitt lymphoma) and OCI (DLBCL) were observed. Curiously, we found that -NH_2_ or -*N*(Me)H sidechain-containing SIRT2 inhibitors while somewhat less- potent on the enzyme in-vitro assay, are more potent in cellular context as compared their polar side chain containing analogs e.g., **55** vs. **46**. While off-target activities cannot be ruled out without further screening, this discrepancy could also arise from differences in cell-permeability or efflux.

We observed that SIRT2 inhibition leads to dose-dependent apoptotic cell death as seen by PARP-cleavage in western blot analysis and in annexin V assay (Figure 3). While uncovering the exact mechanism is beyond the scope of the present study, several hypotheses found in the literature can be invoked. SIRT2 inhibition may correct BCL6 and p53 acetylation imbalance, a known driver of lymphomagenesis [39,47]. Additionally, SIRT2 inhibition has been shown to exert an anti-proliferative effect by facilitating degradation of c-myc; [35] the oncogene which is constitutively active in several Burkitt lymphomas. Blocking of cell proliferation through deactivation of G6PD by SIRT2-mediated deacetylation is another potential mechanism [16]. Pleiotropic effects on other client proteins cannot be ruled out as suggested by several groups in recent studies [45,48]. Finally, recent reports have demonstrated that acyl modifications of lysine residues (e.g., myristoylation) may play a role in the biological functions of the sirtuins. Only a detailed comprehensive evaluation of the myriad acylation/de-acylation activities can shed light on the relevance of the various acyl lysine modifications.

In conclusion, we have developed cambinol analogs, a class of sirtuin inhibitors distinct from other published compounds targeting class III deacylases. Our lead compounds display high SIRT2 inhibitory potency (as low as 250 nM) with excellent selectivity profiles against other sirtuin isoforms. The compounds are cytotoxic to various cancer cell lines, B-cell derived lymphomas (Burkitt lymphoma and DLBCL) in particular. From a medicinal chemistry standpoint, further optimization of the cambinol scaffold with the goal of higher potency and improved physicochemical properties is currently underway and will be reported in due course.

## 4. Materials and Methods

### 4.1. Enzyme Inhibition Assays

SIRT1, SIRT2, and SIRT3 were purchased from Cayman Chemical. Enzyme inhibition assays were performed in 96-well black plates using the SIRT-Glo Assay (Promega Corp.) according to the manufacturer’s instructions. Compounds were dissolved in 100% DMSO and to make 5- or 10-mM stock solutions. For initial enzyme inhibition tests, 50 µM of inhibitory concentration was used. The results are average of triplicate experiments (final DMSO concentration ~0.25%). IC_50_ values were determined in triplicate and reported values are averages of two independent experiments.

### 4.2. Cell Viability Assays

Cell lines were obtained from ATCC (Manassas, VA, USA) and were grown under standard conditions as recommended. For viability assays, 3000 cells per well were dispensed into 96-well plates and treated with test compounds (final DMSO concentration 0.25%). Cells were incubated with test compounds (or DMSO controls) for 72 h or 5 days as noted at 37 °C in 5% CO_2_ and viability was determined using the CellTiter-Glo Assay (Promega Corp.) according to manufacturer’s instructions. Assays were carried out in triplicate and reported values are averages of at least two independent experiments.

### 4.3. General Chemistry Information

Thin layer chromatography was carried out using 60 F254 silica gel plates (Merck, Darmstadt, Germany) using appropriate solvent mixtures. Solvents were ACS reagent grade and anhydrous solvents (Sigma-Aldrich, St. Louis, Missouri, USA, and Acros Organics/Fisher Scientific, Pittsburgh, PA USA,) were used as received. Medium pressure chromatography for compound purification was carried out using Isolera with Silicycle HP cartridges (Biotage, Charlotte, NC, USA,). LCMS was performed using an 1100 HPLC system (Agilent, Santa Clara, CA, USA) equipped with a XTerra MS C18 5 µm, 4.6 Å~ 50 mm column (Waters, Milford, MA, USA ) or a Poroshell 120 EC-C18 4.6 × 100 mm column (Agilent, Santa Clara, CA, USA) using an Agilent photodiode array detector and an in-line Agilent 6130 single quadrupole mass spectrometer. Analytical HPLC method involved gradient elution from 0 to 95% acetonitrile in water (0.1% formic acid) over 6 min. Agilent ChemStation software (Agilent, Santa Clara, CA, USA) was used to develop methods. Final purity of the compounds was determined by ^1^H-NMR (AV-300, Brucker, Billerica, MA, USA) or by analyzing chromatogram of the products at 210, 254 and 280 nm.

### 4.4. Representative Procedure (I) for Preparation of Substituted 2-Hydroxy-1-Naphthalene Carbaldehydes *(**7**)*

*6-Bromo-2-hydroxynaphthalene-1-carbaldehyde* (AstaTech, Bristol, PA, USA). A solution of TiCl_4_ (1.0 M in CH_2_Cl_2_, 27 mmol, 27 mL) in dichloromethane was treated with a solution of dichloromethyl methyl ether (13.5 mmol) in 1,2-dichloroethane or dichloromethane (3 mL) at 0 °C for 15 min. A solution of 6-bromo-2-hydroxy naphthalene (13.5 mmol) in CH_2_Cl_2_ (30 mL) was added dropwise, and the reaction was allowed to stir overnight with gradual warm up to the room temperature. The reaction was quenched by adding 1 M HCl (10 mL). The aqueous layer was extracted with CH_2_Cl_2_ (×3) and the organic layers were then combined, dried over Na_2_SO_4_, and reduced to dryness to afford a reddish-brown residue which was further purified using medium pressure chromatography using a gradient EtOAc/hexane solvent system (1−10% EtOAc) to yield 1.4 g (1.20 g, 5.8 mmol, 43%) of the desired product as a white powder. ^1^H-NMR (300 MHz, DMSO-d_6_) δ 12.01–11.79 (bs, 1H), 10.76 (s, 1H), 8.91 (d, *J* = 9.0 Hz, 1H), 8.17 (d, *J* = 1.5 Hz, 1H), 8.11 (d, *J* = 9.0 Hz, 1H), 7.73 (dd, *J* = 9.0, 1.8 Hz, 1H), 7.30 (d, *J* = 9.0 Hz, 1H). LRMS: *m*/*z* = 248.9 (M − H)^−^.

### 4.5. Representative Procedure (II) for Preparation of Substituted 2-Benzoyl-3H-benzo-[f]chromen-3-ones *(**13**)*

*8-Bromo-2-(3-bromobenzoyl)-3H-benzo[f]chromen-3-one* (**12**). To a solution of substituted 6-bromo-2-hydroxyl naphthaldehyde (124 mg, 0.5 mmol) in ethanol (5 mL) were added the corresponding ethyl-(3-bromophenyl)-3-oxopropanoate 95.5 µL (0.5 mmol). Piperidine (5 drops) was added, and the reaction was heated under reflux for 2 h. The reaction was allowed to cool, and the yellowish precipitate obtained was collected by filtration and washed with ethanol several times to get the condensation product 8-bromo-2-(3-bromobenzoyl)-3H-benzo[f]chromen-3-one, 200 mg (0.20 g, 0.44 mmol, 88%). ^1^H-NMR (300 MHz, DMSO-d_6_) δ 9.24 (s, 1H), 8.59 (d, *J* = 9.0 Hz, 1H), 8.43 (s, 1H), 8.17 (d, *J* = 9.0 Hz, 1H), 8.18 (s, 1H), 7.99 (d, *J* = 7.8 Hz, 1H), 7.90 (t, *J* = 7.5 Hz, 2H), 7.74 (d, *J* = 9.3 Hz, 1H), 7.53 (t, *J* = 7.8 Hz, 1H). LRMS *m*/*z* = 457.0 (M + H)^+^.

### 4.6. Representative Procedure (III) for Preparation of Substituted 2-(Benzoyl)-1H,2H-naphtho [2,1-b]pyran-3-ones *(**14**)*

*8-Bromo-2-(3-bromobenzoyl)-1H,2H-naphtho [2,1-b]pyran-3-one* (**13**). The corresponding benzoyl coumarin (0.20 g, 0.44 mmol) was dissolved in dry pyridine (2 mL). To this solution was added NaBH_4_ (1.25 equiv., 0.55 mmol, 20.9 mg), and the reaction was stirred at room temp for 3 h. The mixture was then poured in cold 2 M hydrochloric acid (5 mL), which resulted in a white precipitate. The precipitate was washed several times with water, dried under a vacuum to yield the corresponding 1,2-dihydrocoumarin, 8-bromo-2-(3-bromobenzoyl)-1H,2H-naphtho[2,1-b]pyran-3-one as a white powder (0.17 g, 0.36 mmol, 82%) which was taken to the next step without purification. ^1^H-NMR (300 MHz, DMSO-d_6_) δ 8.29 (d, *J* = 7.8 Hz, 2H), 8.09 (d, *J* = 7.8 Hz, 1H), 8.03–7.87 (m, 3H), 7.71 (d, *J* = 9.9 Hz, 1H), 7.53 (t, *J* = 7.8 Hz, 1H), 7.42 (d, *J* = 9.0 Hz, 1H), 5.38 (dd, *J* = 11.1, 6.6 Hz, 1H), 3.81-3.54 (ddd, *J* = 16.8, 11.7, 6.6 Hz, 2H). LRMS *m/z* 458.9 (M + H)^+^.

### 4.7. Representative Procedure (IV) for Preparation of Substituted 2-[(2-Hydroxynaphthalen-1-yl)methyl]-3-oxo-3-phenylpropanamides

To a solution of 1,2-dihydrocoumarin, (0.29 mmol) in anhydrous THF (3 mL) was added ammonium hydroxide solution (12 M) 223 µL or the desired amine (1.0 equiv., TEA 2.0 equiv., Sigma-Aldrich, St. Louis, MO, USA) and the reaction was stirred at rt for 8–14 h. Upon completion of the reaction (as judged by TLC/LC-MS), the solution was concentrated in vacuo. The reaction mixture was then purified using HP Silicycle columns using Biotage purification system using Hx/EtOAc gradient elution.

*2-[(6-Bromo-2-hydroxynaphthalen-1-yl)methyl]-3-(4-methylphenyl)-3-oxopropanamide* (**5**). 60.0 mg (0.15 mmol, 50%). ^1^H-NMR (300 MHz, DMSO-d_6_) δ 9.84 (s, 1H), 7.99 (d, *J* = 2.1 Hz, 1H), 7.96–7.85 (m, 3H), 7.64 (d, *J* = 9.0 Hz, 1H), 7.48 (dd, *J* = 9.0, 1.8 Hz, 2H), 7.27 (d, *J* = 8.1 Hz, 2H), 7.20 (d, *J* = 8.7 Hz, 1H), 6.92 (s, 1H), 4.69 (dd, *J* = 8.4, 5.1 Hz, 1H), 3.46 (ddd, *J* = 13.8, 8.4, 5.1 Hz, 2H), 2.35 (s, 3H). LRMS [ES]^+^: *m*/*z* = 412.1 (M + H)^+^.

*2-[(6-Bromo-2-hydroxynaphthalen-1-yl)methyl]-3-(3-methylphenyl)-3-oxopropanamide* (**16**). 6.8 mg (0.02 mmol, 92%). ^1^H-NMR (300 MHz, DMSO-d_6_) δ 9.99-9.75 (bs, 1H), 7.99 (d, *J* = 2.1 Hz, 1H), 7.93 (d, *J* = 9.3 Hz, 1H), 7.79–7.73 (m, 2H), 7.64 (d, *J* = 9.0 Hz, 1H), 7.52–7.46 (m, 2H), 7.44–7.31 (m, 2H), 7.20 (d, *J* = 8.7 Hz, 1H), 6.98–6.88 (bs, 1H), 4.67 (dd, *J* = 8.1, 5.4 Hz, 1H), 3.46 (ddd, *J* = 13.8, 8.7, 5.4 Hz, 2H), 2.32 (s, 3H).LRMS [ES]^+^: *m*/*z* = 412.1 (M + H)^+^.

*2-[(6-Bromo-2-hydroxynaphthalen-1-yl)methyl]-3-(2-methylphenyl)-3-oxopropanamide***(17**). 58.2 mg (0.14 mmol, 93%). ^1^H-NMR (300 MHz, DMSO-d_6_) δ 9.85 (s, 1H), 8.00 (d, *J* = 1.8 Hz, 1H), 7.91 (d, *J* = 9.3 Hz, 1H), 7.65 (d, *J* = 8.7 Hz, 2H), 7.48 (dd, *J* = 9.3, 2.1 Hz, 1H), 7.41–7.32 (m, 2H), 7.27–7.15 (m, 3H), 6.87 (s, 1H), 4.59–4.49 (m, 1H), 3.51–3.41 (m, 2H), 2.29 (s, 3H). LRMS [ES]^+^: *m*/*z* = 412.1 (M + H)^+^.

*2-[(6-Bromo-2-hydroxynaphthalen-1-yl)methyl]-3-oxo-3-phenylpropanamide* (**18**). 13.8 mg (0.03 mmol, 28%). ^1^H-NMR data for the major tautomer is provided. ^1^H-NMR (300 MHz, DMSO-d_6_) δ 9.86 (s, 1H), 8.01–7.95 (m, 2H), 7.92 (d, J = 9.0 Hz, 1H), 7.80 (d, J = 7.2 Hz, 1H), 7.67–7.61 (m, 3H), 7.52–7.43 (m, 3H), 7.20 (d, *J* = 8.7 Hz, 1H), 6.95 (s, 1H), 4.69 (dd, *J* = 8.4, 5.1 Hz, 1H), 3.60–3.38 (m, 2H). LRMS [ES]^+^: *m*/*z* = 398.0 (M + H)^+^.

*2-[(6-Bromo-2-hydroxynaphthalen-1-yl)methyl]-3-(4-chlorophenyl)-3-oxopropanamide* (**19**). 21.8 mg (0.05 mmol, 42%). ^1^H-NMR data for the major isomer is provided. ^1^H-NMR (300 MHz, DMSO-d_6_) δ 9.95- 9.76 (bs, 1H), 8.05-7.86 (m, 4H), 7.64 (d, *J* = 9.0 Hz, 1H), 7.57–7.44 (m, 4H), 7.18 (d, *J* = 8.7 Hz, 1H), 6.97 (s, 1H), 4.66 (dd, *J* = 8.4, 5.1 Hz, 1H), 3.47 (ddd, *J* = 13.8, 8.4, 5.4 Hz, 2H). LRMS [ES]^+^: *m*/*z* = 432.0 (M + H)^+^.

*2-[(6-Bromo-2-hydroxynaphthalen-1-yl)methyl]-3-(4-trifluoromethylphenyl)-3-oxopropanamide* (**20**). 30.8 mg (0.07 mmol, 59%). ^1^H-NMR data for the major isomer is provided. ^1^H-NMR (300 MHz, DMSO-d_6_) δ 9.89 (s, 1H), 8.12 (d, *J* = 8.4 Hz, 1H), 7.99 (d, J = 2.1 Hz, 1H), 7.95–7.80 (m, 4H), 7.64 (d, J = 8.7 Hz, 1H), 7.56–7.45 (m, 2H), 7.18 (d, *J* = 8.7 Hz, 1H), 7.12–6.98 (bs, 1H), 4.72 (dd, *J* = 8.4, 5.4 Hz, 1H), 3.49 (ddd, *J* = 14.4, 8.4, 5.1 Hz, 2H). LRMS [ES]^+^: *m*/*z* = 466.0 (M + H)^+^.

*2-[(6-Bromo-2-hydroxynaphthalen-1-yl)methyl]-3-(3-trifluoromethylphenyl)-3-oxopropanamide* (**21**). 35.5 mg (0.08 mmol, 68%). ^1^H-NMR data for the major isomer is provided. ^1^H-NMR (300 MHz, DMSO-d_6_) δ 9.89 (bs, 1H), 8.22–8.12 m, 1H), 7.98 (d, *J* = 2.1 Hz, 1H), 7.94-7.85 (m, 2H), 7.69 (t, *J* = 7.8 Hz, 1H), 7.62 (d, *J* = 8.7 Hz, 1H), 7.58–7.53 (bs, 1H), 7.51–7.44 (m, 1H), 7.17 (d, *J* = 8.7 Hz, 1H), 6.99 (bs, 1H), 4.75 (dd, *J* = 8.1, 5.7 Hz, 1H), 3.58–3.44 (m, 2H). LRMS [ES]^+^: *m*/*z* = 466.1 (M + H)^+^.

2-[(6-Bromo-2-hydroxynaphthalen-1-yl)methyl]-3-(4-bromo)-3-oxopropanamide (**22**). 21.1 mg (0.02 mmol, 48%). LRMS [ES]^+^: m/z = 478.0 (M + H)^+^.

*2-[(6-Bromo-2-hydroxynaphthalen-1-yl)methyl]-3-(3-bromo)-3-oxopropanamide* (**23**). ^1^H-NMR data for the major isomer is provided. ^1^H-NMR (300 MHz, DMSO-d_6_) δ 9.98–9.87 (bs, 1H), 8.07 (s, 1H), 7.99 (d, *J* = 1.8 Hz, 1H), 7.95–7.86 (m, 2H), 7.82–7.74 (m, 1H), 7.64 (d, *J* = 8.7 Hz, 1H), 7.54–7.39 (m, 3H), 7.19 (d, *J* = 9.0 Hz, 1H), 6.97 (bs, 1H), 4.66 (dd, *J* = 8.1, 5.1 Hz, 1H), 3.57–3.37 (m, 2H). LRMS [ES]^+^: *m*/*z* = 477.0 (M + H)^+^.

*2-[(6-Bromo-2-hydroxynaphthalen-1-yl)methyl]-3-(4-methoxy)-3-oxopropanamide* (**24**). 44.0 mg (0.11 mmol, 85%). ^1^H-NMR (300 MHz, DMSO-d_6_) δ 10.02–9.57 (bs, 1H), 8.02–7.95 (m, 3H), 7.92 (d, *J* = 9 Hz, 1H), 7.64 (d, *J* = 8.7 Hz, 1H), 7.52-7.44 (m, 2H), 7.20 (d, *J* = 9.0 Hz, 1H), 6.98 (d, *J* = 8.7 Hz, 2H), 6.92 (bs, 1H), 4.62 (dd, *J* = 8.4, 5.1 Hz, 1H), 3.82 (s, 3H), 3.38 (ddd, *J* = 13.5, 8.4, 5.1 *Hz*, 2H). LRMS [ES]^+^: *m*/*z* = 428.1 (M + H)^+^.

*2-[(6-Bromo-2-hydroxynaphthalen-1-yl)methyl]-3-(4-fluoro)-3-oxopropanamide* (**25**). 19.5 mg (0.05 mmol, 37%). ^1^H-NMR (300 MHz, DMSO-d_6_) δ 9.87(s, 1H), 8.04 (dd, *J* = 9.0, 5.4 Hz, 2H), 7.99 (d, *J* = 2.1 Hz, 1H), 7.90(d, *J* = 2.3 Hz, 1H), 7.64 (d, *J* = 9.0 Hz, 1H), 7.53–7.44 (m, 1H), 7.30 (t, *J* = 9.0 Hz, 2H), 7.19 (d, *J* = 9.0 Hz, 1H), 7.00–6.93 (bs, 1H), 4.66 (dd, *J* = 8.1, 5.4 Hz, 1H), 3.56–3.41 (m, 2H). LRMS [ES]^+^: *m*/*z* = 416.0 (M + H)^+^.

2-[(6-Bromo-2-hydroxynaphthalen-1-yl)methyl]-3-oxo-3-(pyridin-4-yl)propenamide **(26**). 37.8 mg (0.09 mmol, 46%). LRMS [ES]^+^: m/z = 413.1 (M + H)^+^.

*2-[(6-Bromo-2-hydroxynaphthalen-1-yl)methyl]-3-oxo-3-(pyridin-3-yl)propenamide* (**27**). 12.3 mg (0.03 mmol, 19%). ^1^H-NMR (300 MHz, DMSO-d_6_) δ 9.90 (s, 1H), 9.09–8.98 (bs, 1H), 8.82–8.68 (bs, 1H), 8.19 (td, *J* = 7.8, 1.8 Hz, 1H), 7.99 (d, *J* = 2.1 Hz, 1H), 7.82 (d, *J* = 9.3 Hz, 1H), 7.63 (d, *J* = 8.7 Hz, 1H), 7.54–7.46 (m, 3H), 7.18 (d, *J* = 9.0 *Hz*, 1H), 4.69–4.56 (m, 1H), 3.54–3.42 (m, 2H), 2.33 (d, *J* = 4.5 Hz, 3H). LRMS [ES]_+_: *m*/*z* = 413.0 (M + H)^+^.

*2-[(6-Bromo-2-hydroxynaphthalen-1-yl)methyl]-3-(4-acetamidophenyl)-3-oxopropanamide* (**28**). 14.0 mg (0.03 mmol, 26%). ^1^H-NMR (300 MHz, DMSO-d_6_) δ 10.09 (s, 1H), 9.86 (s, 1H), 8.11–8.05 (bs, 1H), 7.99 (d, *J* = 2.1 Hz, 1H), 7.91–7.78 (m, 3H), 7.64 (d, *J* = 8.7 Hz, 2H), 7.49 (dd, *J* = 9.0, 2.1 Hz, 1H), 7.37 (d, *J* = 8.1 Hz, 1H), 7.19 (d, *J = 8.7 Hz*, 1H), 4.63–4.50 (m, 1H), 3.54–3.43 (m, 2H), 2.31 (d, *J* = 4.5 Hz, 3H), 2.05 (s, 3H). LRMS [ES]^+^: *m*/*z* = 469.0 (M + H)^+^.

*3-(3-Bromophenyl)-2-[(2-hydroxynaphthalen-1-yl)methyl]-3-oxopropanamide* (**29**). 48.5 mg (0.12 mmol, 93%). ^1^H-NMR data for the major isomer is provided. ^1^H-NMR (300 MHz, DMSO-d_6_) δ 9.65 (s, 1H), 8.06 (t, *J* = 1.5 Hz, 1H), 7.92 (d, *J* = 8.4 Hz, 1H), 7.83–7.72 (m, 2H), 7.63 (d, *J* = 9.0 Hz, 1H), 7.57–7.52 (bs, 1H), 7.46–7.34 (m, 3H), 7.28–7.22 (m, 1H), 7.14 (d, *J* = 8.7 Hz, 1H), 7.04–6.97 (bs, 1H), 4.70 (dd, *J* = 8.1, 5.4 Hz, 1H), 3.49 (ddd, *J* = 13.8, 8.1, 5.4 Hz, 2H). LRMS [ES]^+^: *m*/*z* = 398.0 (M + H)^+^.

3-(3-Bromophenyl)-2-[(2-hydroxy-6-methoxynaphthalen-1-yl)methyl]-3-oxopropanamide (**30**). 15.3 mg (0.36 mmol, 92%). LRMS [ES]^+^: m/z = 410.0 (M + H)^+^.

*2-[(3-Bromo-2-hydroxynaphthalen-1-yl)methyl]-3-(3-bromophenyl)-3-oxopropanamide* (**31**). 27.2 mg (0.06 mmol, 44%). ^1^H-NMR data for the major isomer is provided. ^1^H-NMR (300 MHz, DMSO-d_6_) δ 9.98 (s, 1H), 8.14-8.08 (m, 2H), 7.99–7.92 (m, 1H), 7.83–7.66 (m, 3H), 7.63–7.53 (m, 1H), 7.47–7.32 (m, 2H), 7.25–7.13 (m, 1H), 7.12–6.97 (m, 1H), 4.78–4.63 (m, 1H), 3.56–3.45 (m, 2H). LRMS [ES]^+^: *m*/*z* = 476.00 (M + H)^+^.

2-[(7-Bromo-2-hydroxynaphthalen-1-yl)methyl]-3-(3-bromophenyl)-3-oxopropanamide (**32**). 61.7 mg (0.13 mmol, 100%). LRMS [ES]^+^: m/z = 476.00 (M + H)^+^.

*2-[(6-Bromo-2-hydroxynaphthalen-1-yl)methyl]-N-methyl-3-oxo-3-[4-(trifluoromethyl)phenyl]propenamide* (**33**). 33.2 mg (0.07 mmol, 69%). ^1^H-NMR data for the major isomer is provided. ^1^H-NMR (300 MHz, DMSO-d_6_) δ 9.94–9.80 (bs, 1H), 8.08 (d, *J* = 8.1 Hz, 1H), 7.99 (d, *J* = 1.8 Hz, 1H), 7.98–7.90 (m, 1H), 7.87–7.75 (m, 4H), 7.67–7.61 (m, 1H), 7.52–7.46 (m, 1H), 7.18 (d, *J* = 9.0 Hz, 1H), 4.64 (t, *J* = 7.2 Hz, 1H), 3.48 (d, *J* = 6.3 Hz, 2H), 2.31 (d, *J* = 4.5 Hz, 3H).

*2-[(2-Hydroxy-6-methylnaphthalen-1-yl)methyl]-3-oxo-3-[4-(trifluoromethyl)phenyl]propenamide* (**34**). 48.0 mg (0.11 mmol, 70%). ^1^H-NMR data for the major isomer is provided. ^1^H-NMR (300 MHz, DMSO-d_6_) δ 9.59–9.40 (bs, 1H), 8.07 (d, *J* = 8.4 Hz, 2H), 7.99–7.89 (m, 1H), 7.86–7.72 (m, 2H), 7.67–7.61 (m, 1H), 7.56–7.47 (m, 2H), 7.44–7.38 (m, 1H), 7.23 (bd, *J* = 8.7 Hz, 1H), 7.08 (d, *J* = 8.7 Hz, 1H), 4.72–4.59 (m, 1H), 3.53–3.45 (m, 2H), 2.39 (s, 3H), 2.33 (d, *J* = 4.5 Hz, 3H). LRMS [ES]^+^: *m*/*z* = 416.1 (M + H)^+^.

*2-[(2-Chloro-6-hydroxyquinolin-5-yl)methyl]-3-oxo-3-[4-(trifluoromethyl)phenyl]propenamide* (**35**). 41.2 mg (0.09 mmol, 75%). ^1^H-NMR data for the major isomer is provided. ^1^H-NMR (300 MHz, DMSO-d_6_) δ 10.28–10.10 (bs, 1H), 8.35 (d, *J* = 9.0 Hz, 1H), 8.10 (d, *J* = 8.4 Hz, 2H), 8.02–7.92 (m, 1H), 7.85 (d, *J* = 8.4 Hz, 2H), 7.71 (d, *J* = 9.0 Hz, 1H), 7.44 (dd, *J* = 9.6, 7.2 Hz, 2H), 4.74–4.57 (m, 1H), 3.53–3.42 (m, 2H), 2.28 (d, *J* = 4.5 Hz, 3H). LRMS [ES]^+^: *m*/*z* = 437.1 (M + H)^+^.

*2-[(6-Bromo-2-hydroxynaphthalen-1-yl)methyl]-3-(3-bromophenyl)-N-(2-hydroxyethyl)-3-oxopropanamide* (**36**). 4.7 mg (0.01 mmol, 16%). ^1^H-NMR data for the major isomer is provided. ^1^H-NMR (300 MHz, DMSO-d_6_) δ 10.03-9.49 (bs, 1H), 8.04 (s, 1H), 7.99 (s, 1H), 7.89 (d, *J* = 7.5 Hz, 1H), 7.85–7.74 (m, 2H), 7.63 (d, *J* = 9.0 Hz, 1H), 7.44 (d, *J* = 9.3 Hz, 1H), 7.40 (t, *J* = 7.8 Hz, 1H), 7.18 (d, *J* = 9.0 Hz, 2H), 4.71–4.57 (m, 1H), 3.50–3.41 (m, 2H), 3.13–2.79 (m, 4H). LRMS [ES]^+^: *m*/*z* = 520.0 (M + H)^+^.

*2-[(6-Bromo-2-hydroxynaphthalen-1-yl)methyl]-3-(3-bromophenyl)-N-(3-hydroxypropyl)-3-oxopropanamide* (**37**). 10.4 mg (0.02 mmol, 21%). ^1^H-NMR data for the major isomer is provided. ^1^H-NMR (300 MHz, DMSO-d_6_) δ 8.04 (s, 1H), 7.99 (bs, 1H), 8.10 (d, *J* = 7.8 Hz, 1H), 7.80 (d, *J* = 9.0 Hz, 2H), 7.68–7.57 (m, 2H), 7.52–7.38 (m, 3H), 7.20 (d, *J* = 9.0 Hz, 1H), 4.66–4.53 (m, 1H), 3.56–3.39 (m, 2H), 3.11–2.69 (m, 4H), 1.20–1.03 (m, 2H). LRMS [ES]^+^: *m*/*z* = 434.0 (M + H)^+^.

*2-[(6-Bromo-2-hydroxynaphthalen-1-yl)methyl]-1-(3-bromophenyl)-3-(morpholin-4-yl)propane-1,3-dione* (**38**). 25.9 mg (0.05 mmol, 52%). ^1^H-NMR data for the major isomer is provided. ^1^H-NMR (300 MHz, DMSO-d_6_) δ 10.04 (s, 1H), 8.05 (d, *J = 1.8 Hz*, 1H), 7.90 (t, *J* = 1.8 Hz, 1H), 7.83–7.68 (m, 4H), 7.51 (dd, *J* = 9.3, 2.1 Hz, 1H), 7.45 (t, *J* = 7.8 Hz, 1H), 7.22 (d, *J* = 8.7 Hz, 1H), 5.00 (dd, *J* = 9.3, 5.4 Hz, 1H), 3.60–3.45 (m, 2H), 3.31–2.91 (m, 6H), 2.87–2.76 (m, 1H), 2.45–2.35 (m,1H). LRMS [ES]^+^: *m*/*z* = 546.0 (M + H)^+^.

*2-[(6-Bromo-2-hydroxynaphthalen-1-yl)methyl]-3-(3-bromophenyl)-N-(furan-3-ylmethyl)-3-oxopropanamide* (**39**). 34.4 mg (0.08 mmol, 17%). ^1^H-NMR data for the major isomer is provided. ^1^H-NMR (300 MHz, DMSO-d_6_) δ 9.89 (s, 1H), 8.51 (t, *J = 5.4 Hz*, 1H), 8.04 (t, *J* = 1.5 Hz, 1H), 7.99 (d, *J* = 1.8 Hz, 1H), 7.88 (d, *J* = 7.8 Hz, 1H), 7.84–7.74 (m, 2H), 7.63 (d, *J* = 8.7 Hz, 1H), 7.49–7.34 (m, 3H), 7.18 (d, *J* = 9.0 Hz, 1H), 6.23 (dd, *J* = 3.0, 1.8 Hz, 1H), 5.76 (d, *J* = 2.7 Hz, 1H), 4.70 (t, *J* = 6.0 Hz, 1H), 4.00 (dd, *J* = 15.6, 5.4, 5.1 *Hz*, 2H), 3.55–3.44 (m, 2H). LRMS [ES]^+^: *m*/*z* = 556 (M + H)^+^.

*2-[(6-Bromo-2-hydroxynaphthalen-1-yl)methyl]-3-(3-bromophenyl)-3-oxo-N-(pyridin-4-ylmethyl)-propenamide* (**40**). 30.7 mg (0.05 mmol, 31%). ^1^H-NMR data for the major isomer is provided. ^1^H-NMR (300 MHz, DMSO-d_6_) δ 9.94 (bs, 1H), 8.73-8.60 (m, 1H), 8.24 (d, *J* = 5.1 Hz, 2H), 8.13 (s, 1H), 8.05 (s, 1H), 7.94 (d, *J* = 7.8 Hz, 1H), 7.84 (t, *J* = 9.3 Hz, 2H), 7.70 (d, *J* = 8.7 Hz, 1H), 7.51−7.41 (m, 2H), 7.23 (d, *J* = 9.0 Hz, 1H), 6.66 (d, *J* = 4.8 Hz, 2H), 4.87−4.72 (m, 1H), 4.06−3.98 (m, 2H), 3.69−3.38 (m, 2H). LRMS [ES]^+^: *m*/*z* = 567.0 (M + H)^+^.

*2-[(6-Bromo-2-hydroxynaphthalen-1-yl)methyl]-3-(3-bromophenyl)-N-(2-methoxyethyl)-3-oxopropanamide* (**41**). 25.0 mg (0.47 mmol, 31%). ^1^H-NMR data for the major isomer is provided. ^1^H-NMR (300 MHz, DMSO-d_6_) δ 9.91–9.75 (bs, 1H), 8.19–8.06 (m, 2H), 8.03–7.98 (m, 1H), 7.93 (d, *J* = 7.8 Hz, 1H), 7.83–7.74 (m, 2H), 7.64 (d, *J* = 8.7 Hz, 1H), 7.48 (bd, *J = 9.3 Hz*, 1H), 7.41 (t, *J* = 7.8 Hz, 1H), 7.20 (d, *J* = 9.0 Hz, 1H), 4.64 (dd, *J* = 8.4, 4.5 Hz, 1H), 3.44 (ddd, *J* = 13.8, 8.4, 4.5 Hz, 2H), 3.03–2.82 (m, 6H), 2.77–2.64 (m, 1H). LRMS [ES]^+^: *m*/*z* = 534.1 (M + H)^+^.

*2-[(6-Bromo-2-hydroxynaphthalen-1-yl)methyl]-3-(3-bromophenyl)-3-oxo-N-(pyridin-3-ylmethyl)-propenamide* (**42**). 35.2 mg (0.06 mmol, 41%). ^1^H-NMR data for the major isomer is provided. ^1^H-NMR (300 MHz, DMSO-d_6_) δ 9.90 (bs, 1H), 8.59 (t, *J* = 5.1 Hz, 1H), 8.38–8.32 (m, 1H), 8.20-8.13 (m, 1H), 8.07 (t, *J* = 1.5 Hz, 1H), 8.01 (d, *J* = 2.1 Hz, 1H), 7.93-7.77 (m, 2H), 7.66 (d, *J* = 8.7 Hz, 1H), 7.46 (dd, *J* = 9.0, 2.1 Hz, 1H), 7.40 (t, *J* = 7.8 Hz, 1H), 7.20 (d, *J* = 8.7 Hz, 1H), 7.10−7.04 (m, 1H), 7.02−6.95 (m, 1H), 4.70 (dd, *J* = 9.0, 4.5 Hz, 1H), 4.05−3.98 (m, 2H), 3.62−3.38 (m, 2H). LRMS [ES]^+^: *m*/*z* = 567.0 (M + H)^+^.

*2-[(6-Bromo-2-hydroxynaphthalen-1-yl)methyl]-3-(3-bromophenyl)-N-(2-carbamoylethyl)-3-oxopropanamide* (**43**). 35.4 mg (0.06 mmol, 31%). ^1^H-NMR data for the major isomer is provided. ^1^H-NMR (300 MHz, DMSO-d_6_) δ 9.84 (s, 1H), 8.12–8.04 (m, 1H), 8.03–7.98 (m, 2H), 7.87 (d, *J* = 8.1 Hz, 1H), 7.783–7.74 (m, 2H), 7.63 (d, *J* = 9.0 Hz, 1H), 7.53–7.47 (m, 1H), 7.39 (t, *J* = 8.1 Hz, 1H), 7.17 (d, *J* = 8.7 Hz, 1H), 7.15−7.09 (bs, 1H), 6.77−6.67 (bs, 1H), 4.66–4.55 (m, 1H), 3.50−3.41 (m, 2H), 3.08−2.84 (m, 2H), 1.92 (dt, *J = 7.2, 2.1 Hz*, 2H).

*2-[(6-Bromo-2-hydroxynaphthalen-1-yl)methyl]-3-(3-bromophenyl)-N-[2-(methylcarbamoyl)ethyl]-3-oxo-propanamide* (**44**). 22.1 mg (0.04 mmol, 30%). ^1^H-NMR data for major isomers are provided. ^1^H-NMR (300 MHz, DMSO-d_6_) δ 9.84 (d, *J = 5.7 Hz*, 1H), 8.12–7.96 (m, 3H), 7.94–7.69 (m, 3H), 7.68–7.45 (m, 3H), 7.44–7.33 (m, 1H), 7.22–7.13 (m, 1H), 4.68–4.54 (m, 1H), 3.52–3.41 (m, 2H), 3.08–2.84 (m, 2H), 2.59–2.45 (m, 3H), 1.98–1.83 (m, 2H). LRMS [ES]^+^: *m*/*z* = 561.0 (M + H)^+^.

*2-[(6-Bromo-2-hydroxynaphthalen-1-yl)methyl]-1-(4-methylpiperazin-1-yl)-3-[4-(trifluoromethyl)phenyl]-propane-1,3-dione* (**45**). 30.1 mg (0.06 mmol, 49%). ^1^H-NMR (300 MHz, DMSO-d_6_) δ 10.13-1.04 (bs,1H), 8.05 (d, *J* = 2.1 Hz, 1H), 7.99–7.93 (m, 2H), 7.90–7.80 (m, 3H), 7.78–7.68 (m, 2H), 7.50 (dd, *J* = 9.0, 2.1 *Hz*, 1H), 7.23 (d, *J* = 9.0 Hz, 1H)), 5.12–4.97 (m, 1H), 3.64–3.46 (m, 2H), 3.23–2.93 (m, 3H), 2.72–2.58 (m, 1H), 2.03–1.88 (m, 1H), 1.86 (s, 3H), 1.40–1.23 (m, 1H), 0.92–0.78 (m, 1H). LRMS [ES]^+^: *m*/*z* = 549.1 (M + H)^+^.

2-[(6-Bromo-2-hydroxynaphthalen-1-yl)methyl]-1-[4-(trifluoromethyl)phenyl]hex-5-yne-1,3-dione (**46**). 42.2 mg (0.08 mmol, 42%). LRMS [ES]^+^: m/z = 504.1 (M+H)^+^.

*2-[(6-Bromo-2-hydroxynaphthalen-1-yl)methyl]-3-(4-chlorophenyl)-N-(2-hydroxyethyl)-3-oxopropanamide* (**47**)/ 13.1 mg (0.03 mmol, 14%). ^1^H-NMR (300 MHz, DMSO-d_6_) δ 9.95–9.75 (bs,1H), 7.99 (d, *J* = 2.1 Hz, 2H), 7.96 (d, *J* = 8.4 Hz, 2H), 7.82 (d, *J* = 9.3 Hz, 1H), 7.64 (d, *J* = 9.0 Hz, 1H), 7.52 (d, *J* = 8.7 Hz, 2H), 7.48 (dd, *J* = 9.0, 2.1 Hz, 1H), 7.18 (d, *J* = 8.7 Hz, 1H), 4.67–4.58 (m, 1H), 4.47–4.38 (m, 1H), 3.50–3.42 (m, 2H), 3.10-2.80 (m, 3H). LRMS [ES]^+^: *m*/*z* = 476.0 (M + H)^+^.

2-[(6-Bromo-2-hydroxynaphthalen-1-yl)methyl]-3-(4-bromophenyl)-N-(2-hydroxyethyl)-3-oxopropanamide (**48**). 23.9 mg (0.05 mmol, 25%). ^1^H-NMR (300 MHz, DMSO-d_6_) δ 9.90–9.80 (s,1H), 7.99 (d, J = 2.1 Hz, 2H), 7.87 (d, J = 8.4 Hz, 2H), 7.81 (d, J = 9.0 Hz, 1H), 7.70–7.60 (m, 3H), 7.48 (dd, J = 9.3, 2.1 Hz, 1H), 7.18 (d, J = 8.7 Hz, 1H), 4.68–4.58 (m, 1H), 4.48–4.36 (m, 1H), 3.50–3.42 (m, 2H), 3.11–2.78 (m, 3H). LRMS [ES]^+^: m/z = 520.0 (M + H)^+^.

2-[(6-Bromo-2-hydroxynaphthalen-1-yl)methyl]-N-(2-hydroxyethyl)-3-oxo-3-[4-(trifluoromethyl)phenyl]-propenamide (**49**). 37.9 mg (0.07mmol, 41%). LRMS [ES]^+^: m/z = 510.0 (M + H)^+^.

*2-[(6-Bromo-2-hydroxynaphthalen-1-yl)methyl]-N-(2-carbamoylethyl)-3-(4-chlorophenyl)-3-oxopropanamide* (**50**). 36.0 mg (0.07 mmol, 42%). ^1^H-NMR data for major isomers are provided. ^1^H-NMR (300 MHz, DMSO-d_6_) δ 9.81 (s, 1H), 8.04 (t, *J* = 5.7 Hz, 1H), 7.99 (d, *J* = 1.8 Hz, 1H), 7.92 (d, *J* = 8.7 Hz, 2H), 7.80 (d, *J* = 9.3 Hz, 1H), 7.59 (d, *J* = 9.0 Hz, 1H), 7.55–7.49 (m, 3H), 7.17 (d, *J* = 8.7 Hz, 1H), 7.14–7.06 (bs, 1H), 6.78–6.68 (bs, 1H), 4.63–4.55 (m, 1H), 3.51–3.40 (m, 2H), 3.10–2.81 (m, 2H), 1.96–1.86 (m, 2H). LRMS [ES]^+^: *m*/*z* = 503.1 (M + H)^+^.

*2-[(6-Bromo-2-hydroxynaphthalen-1-yl)methyl]-3-(4-bromophenyl)-N-(2-carbamoylethyl)-3-oxopropanamide* (**51**). 24.1 mg (0.04 mmol, 26%). ^1^H-NMR data for major isomers are provided. ^1^H-NMR (300 MHz, DMSO-d_6_) δ 9.82 (s, 1H), 8.08–8.01 (m, 1H), 7.99 (d, *J* = 1.8 Hz, 1H), 7.84 (d, *J* = 8.7 Hz, 2H), 7.79 (d, *J* = 9.3 Hz, 1H), 7.69–7.60 (m, 3H), 7.49 (dd, *J* = 9.0, 1.8 Hz, 1H), 7.17 (d, *J* = 9.0 Hz, 1H), 7.13–7.08 (bs, 1H), 6.77–6.68 (bs, 1H), 4.64–4.50 (m, 1H), 3.50–3.42 (m, 2H), 3.08–2.81 (m, 2H), 1.95–1.85 (m, 2H). LRMS [ES]^+^: *m*/*z* = 547.0 (M + H)^+^.

*2-[(6-Bromo-2-hydroxynaphthalen-1-yl)methyl]-N-(2-carbamoylethyl)-3-oxo-3-[4-(trifluoromethyl)phenyl]-propenamide* (**52**). 46.3 mg (0.09 mmol, 48%). _1_H-NMR (300 MHz, DMSO-d_6_) δ 9.89-9.74 (bs,1H), 8.07 (d, *J* = 8.4 Hz, 2H), 7.99 (d, *J* = 1.8 Hz, 1H), 7.84–7.74 (m, 3H), 7.68–7.58 (m, 2H), 7.49 (dd, *J* = 9.0, 2.1 *Hz*, 1H), 7.18 (dd, *J* = 9.0, 2.1 Hz, 1H),7.13–7.05 (bs, 1H), 6.77–6.67 (bs, 1H), 4.74–4.57 (m, 1H), 3.52–3.41 (m, 2H), 3.08–2.83 (m, 2H), 1.95–1.86 (m, 2H). LRMS [ES]^+^: *m*/*z* = 537.1 (M + H)^+^.

*2-[(6-Bromo-2-hydroxynaphthalen-1-yl)methyl]-N-(2-carbamoylethyl)-3-oxo-3-[3-(trifluoromethyl)phenyl]-propenamide* (**53**). 25.7 mg (0.03 mmol, 16%). ^1^H-NMR data for major isomers are provided. ^1^H-NMR (300 MHz, DMSO-d_6_) δ 9.83 (s,1H), 8.19–8.07 (m, 2H), 7.98 (d, *J* = 1.8 Hz, 1H), 7.95–7.55 (m, 5H), 7.50 (dd, *J* = 9.0, 2.1 Hz, 1H), 7.18 (d, *J* = 9.0 Hz, 1H), 7.13–7.05 (bs, 1H), 6.76–6.67 (m, 1H), 4.74–4.62 (m, 1H), 3.52–3.42 (m, 2H), 3.12–2.85 (m, 2H), 1.97–1.87 (m, 2H).

*2-[(6-Bromo-2-hydroxynaphthalen-1-yl)methyl]-3-(4-chlorophenyl)-N-[2-(methylcarbamoyl)ethyl]-3-oxo-propanamide* (**54**). 35.8 mg (0.07 mmol, 64%). ^1^H-NMR (300 MHz, DMSO-d_6_) δ 9.92–9.72 (bs,1H), 8.10–7.87 (m, 4H), 7.87–7.76 (m, 1H), 7.70–7.60 (m, 1H), 7.56–7.43 (m, 4H), 7.26–7.11 (m, 1H), 4.66–4.52 (m, 1H), 3.53–3.40 (m, 2H), 3.10–2.82 (m, 2H), 2.55–2.45 (m, 3H), 1.98–1.84 (m, 2H). LRMS [ES]^+^: *m*/*z* = 517.1 (M + H)^+^.

*2-[(6-Bromo-2-hydroxynaphthalen-1-yl)methyl]-N-[2-(methylcarbamoyl)ethyl]-3-oxo-3-[4-(trifluoromethyl)-phenyl]propenamide* (**55**). 60.4 mg (0.12 mmol, 68%). ^1^H-NMR data for major isomers are provided. ^1^H-NMR (300 MHz, DMSO-d_6_) δ 9.80–9.74 (bs,1H), 8.08 (d, *J* = 7.8 Hz, 2H), 7.99 (d, *J* = 1.8 Hz, 1H), 7.80–7.77 (m, 3H), 7.64 (d, *J* = 9.0 Hz, 1H), 7.53–7.45 (m, 2H), 7.17 (d, *J* = 9.0 Hz, 1H), 4.65 (t, *J* = 6.6 Hz, 1H), 3.47 (d, *J* = 6.6 Hz, 2H), 3.18–2.84 (m, 2H), 2.46 (d, *J* = 4.8 Hz, 3H), 1.95–1.84 (m, 2H). LRMS [ES]^+^: *m*/*z* = 551.1 (M + H)_+_.

*2-[(6-Bromo-2-hydroxynaphthalen-1-yl)methyl]-3-(4-bromophenyl)-N-methyl-3-oxopropanamide* (**56**). 51.5 mg (0.11 mmol, 96%). ^1^H-NMR data for major isomers are provided. ^1^H-NMR (300 MHz, DMSO-d_6_) δ 9.98–9.38 (bs,1H), 7.99 (d, *J* = 1.5 Hz, 1H), 7.94–7.79 (m, 3H), 7.72–7.61 (m, 3H), 7.49 (dd, *J* = 9.3, 1.8 Hz, 1H), 7.19 (d, *J* = 8.7 Hz, 1H), 4.64-4.50 (t, *J* = 6.6 Hz, 1H), 3.52–3.41 (m, 2H), 2.31 (d, *J* = 4.2 Hz, 3H).

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
