# Peer review of "Discovery of Selective SIRT2 Inhibitors as Therapeutic Agents in B-Cell Lymphoma and Other Malignancies"

_molecules, 2020, doi:10.3390/molecules25030455_

Round 1

Reviewer 1 Report

I thank the authors for your exhaustive responses

Author Response

We thank reviewer #1 for their positive comment.

Reviewer 2 Report

In the manuscript entitled "Discovery of selective SIRT2 inhibitors as therapeutic agents in B-cell lymphoma and other malignancies", Chowdhury et al described the development of selective SIRT2 inhibitor through multiple rounds of optimizations and showed these inhibitors can induce cytotoxicity in multiple cancer cell lines. The importance of developing selective SIRT2 inhibitors were very obvious for their implications of anti-tumor activity. Although several compounds were reported to be selectively against SIRT2, new inhibitors with optimized structure and functions are always desired. However, with the data in the manuscript, I still have several concerns.

Major concerns:
1. Although the new synthesized inhibitors act much more against SIRT2 than SIRT1 and SIRT3, there remains unknown that these inhibitors might also inhibit closely-related enzymes such as zinc-dependent histone deacetylases (HDACs) including HDAC1 and HDAC6. Such analysis is critical to show the selectivity of these inhibitors. Another informative assay is to determine how SIRT2 overexpression affect the effect of the inhibitors.
2. The mechanism underlying the cytotoxicity of these SIRT2 inhibitors were not determined. Do they induce Caspase dependent or independent apoptotic pathways? What are the intermediate downstream effector genes upon the inhibition of SIRT2?
3. There are a few so-called selective SIRT2 inhibitor reported, including AGK2, SirReal2, Tenovin-6, and TM. The authors mentions some of them here and there. It is still worth mentioning the difference between the newly developed ones with these reported ones at least in the discussion sections.

Minor comments:
- Figure2 and Figure 3E, a quantification result of the Western blot analysis is needed to provide more detailed information on the effects. Also the authors need to provide a summary of all the replicate experiments.
- Figure 3, panel A-D is difficult to read because of the small font size. Also percentage of cells in each gate should be provided.
- Line 66, the expression "2 >50 µM" is confusing.
- The nomenclature of the compounds is confusing as the author used the number with bold font. I suggest the authors add a prefix to the compound to make it a bit more distinguishable and meaningful.

Author Response

Reviewer 2

In the manuscript entitled "Discovery of selective SIRT2 inhibitors as therapeutic agents in B-cell lymphoma and other malignancies", Chowdhury et al described the development of selective SIRT2 inhibitor through multiple rounds of optimizations and showed these inhibitors can induce cytotoxicity in multiple cancer cell lines. The importance of developing selective SIRT2 inhibitors were very obvious for their implications of anti-tumor activity. Although several compounds were reported to be selectively against SIRT2, new inhibitors with optimized structure and functions are always desired. However, with the data in the manuscript, I still have several concerns.

Major concerns: 
1. Although the new synthesized inhibitors act much more against SIRT2 than SIRT1 and SIRT3, there remains unknown that these inhibitors might also inhibit closely-related enzymes such as zinc-dependent histone deacetylases (HDACs) including HDAC1 and HDAC6. Such analysis is critical to show the selectivity of these inhibitors. Another informative assay is to determine how SIRT2 overexpression affect the effect of the inhibitors. 

HDAC1 and HDAC6 are zinc-dependent (class I and II) deacetylase that act by a different mechanism than sirtuins and have no discernable sequence homology to the sirtuins and therefore are not expected to be affected by sirtuin inhibitors.  We previously showed that cambinol has no activity against HDAC1 and HDAC6.  The manuscript has been revised to include this information. 

The mechanism underlying the cytotoxicity of these SIRT2 inhibitors were not determined. Do they induce Caspase dependent or independent apoptotic pathways? What are the intermediate downstream effector genes upon the inhibition of SIRT2?

We present cleavage of PARP (Figure 3) as evidence of caspase-dependent apoptosis.  More exhaustive mechanistic studies are beyond the scope of the current manuscript.

There are a few so-called selective SIRT2 inhibitor reported, including AGK2, SirReal2, Tenovin-6, and TM. The authors mentions some of them here and there. It is still worth mentioning the difference between the newly developed ones with these reported ones at least in the discussion sections.

The discussion section has been revised to include this information.

Minor comments:
- Figure2 and Figure 3E, a quantification result of the Western blot analysis is needed to provide more detailed information on the effects. Also the authors need to provide a summary of all the replicate experiments. 

The quantification has been included.

- Figure 3, panel A-D is difficult to read because of the small font size. Also percentage of cells in each gate should be provided. 

Figures have been revised.

- Line 66, the expression "2 >50 µM" is confusing. 

Revised to make clear.

- The nomenclature of the compounds is confusing as the author used the number with bold font. I suggest the authors add a prefix to the compound to make it a bit more distinguishable and meaningful.

We are unsure what the reviewer means.  Compound numbers are bolded to differentiate them from any other number in the text (e.g. IC50 value).  Adding a prefix would add little to clarify this.

Reviewer 3 Report

In this manuscript, Chowdhury et al. report a series of cambinol-based sirtuin inhibitors. The cytotoxicity toward lymphoma cell lines were tested by measuring the cellular ATP levels. Some of these inhibitors showed increased specificity to SIRT2. The authors conclude that these compounds were cytotoxic against B-cell lymphoma and epithelial cancer cell lines with excellent selectivity profile against sirtuin isoforms in vitro.

Overall, this is a well-designed project and is important to the cancer filed as SIRT2 can be a potential anti-tumor therapy target.

Although it is known from previous studies that the inhibition of sirtunis is well tolerated in normal cells, the specifities of these new inhibitors to sirtunis have not been documented. To rule out the off-target effects of these inhibitors in cancer and lymphoma cells, it is necessary to use normal cells as control to test cell viability with these inhibitors. Data for BrdU and annexin V staining is not provided in the manuscript as indicated in Figure 4 in the text. Methods for flow cytometry for figure 3 is not provided in the manuscript.

Author Response

In this manuscript, Chowdhury et al. report a series of cambinol-based sirtuin inhibitors. The cytotoxicity toward lymphoma cell lines were tested by measuring the cellular ATP levels. Some of these inhibitors showed increased specificity to SIRT2. The authors conclude that these compounds were cytotoxic against B-cell lymphoma and epithelial cancer cell lines with excellent selectivity profile against sirtuin isoforms in vitro.

Overall, this is a well-designed project and is important to the cancer filed as SIRT2 can be a potential anti-tumor therapy target.

Although it is known from previous studies that the inhibition of sirtunis is well tolerated in normal cells, the specifities of these new inhibitors to sirtunis have not been documented. To rule out the off-target effects of these inhibitors in cancer and lymphoma cells, it is necessary to use normal cells as control to test cell viability with these inhibitors. Data for BrdU and annexin V staining is not provided in the manuscript as indicated in Figure 4 in the text. Methods for flow cytometry for figure 3 is not provided in the manuscript. 

The reference to BrdU staining has been removed.  This was part of an earlier version of the manuscript.

Although it is known from previous studies that the inhibition of sirtunis is well tolerated in normal cells, the specifities of these new inhibitors to sirtunis have not been documented. To rule out the off-target effects of these inhibitors in cancer and lymphoma cells, it is necessary to use normal cells as control to test cell viability with these inhibitors. Data for BrdU and annexin V staining is not provided in the manuscript as indicated in Figure 4 in the text. Methods for flow cytometry for figure 3 is not provided in the manuscript

Methods for Annexin V staining are now referenced (done as in ref. 39).

Round 2

Reviewer 2 Report

The authors have addressed all of my concerns. I have no further comments. Thanks. 

This manuscript is a resubmission of an earlier submission. The following is a list of the peer review reports and author responses from that submission.

Round 1

Reviewer 1 Report

The authors should test a negative control (e.g. 32) in proliferation and western blotting. Also a selective and potent Sirt2 inhibitor from the literature with a different scaffold should be compared as well.

The Lin group has shown that inhibition of demyristoylation is important for anticancer activity. Hence it would be good to compare the effect of the inhibitors also on this. The editor should give advice if this is mandatory. it should at least be discussed.

Reviewer 2 Report

In this manuscript the authors identified several SIRT inhibitors, evaluating the most effectives and specifics for SIRT2 and then analyzed its biological activity. The topic is interesting but there are some criticisms.

In general the authors have evaluate different SURT inhibitors. The two most interesting inhibitors are 55 and 56. 55 inhibitor presents a very high affinity for SIRT2 (100), none for SIRT1 (0.0) e very low for SIT3 (20), moreover IC50 is 0.25 and is very efficacy in term of cytotoxicity against different cell line. 56 inhibitor presents affinity for SIRT2 (95) but also affinity for SIRT1 and SIRT3 (although low), and their cytostoxicity are very efficacy against cell lines. In my opinion 55 inhibitor would give more merit to further study compared to 56. I would prefer the specificity with respect to efficacy, furthemore the difference in efficacy between the two inhibitors is minimal. The authors prefer instead to perform experiments on vitality and apoptosis on 55. Please explain the reason.

- In the abstract the authors state “ In particular, compounds 55 (IC50 SIRT2 0.25 mM and >500 mM against SIRT1 and SIRT3) and 56 (IC50 SIRT2 0.78 mM and >250 mM against SIRT1 and >150 mM against SIRT3) showed apoptotic as well as strong anti-proliferative properties against B-cell lymphoma cells” Data about IC50 SIRT2 are reported in Table 5 but data about SIRT1 and SIRT3 are not reported in any table. Furthermore the authors perform apoptotis and antiproliferative experiments only for 56 and not 55 (see previous Comments). Please correct

- Please what correspond AGK2?  

- Section Results, paragraph “Biological evaluation of sirtuin 2 inhibitors”, why the authors have chosen cell line NCI-H460 and 18h for the treatment? Please explain.

- Table V: there are no reported any data about 46 and 57 ( as mentioned in text)

- Please specify that in table V the authors reported the percentage of cell survival (otherwise it doesn't make sense).

- The figure 4 a-b-c-d-e mentioned in text of paragraph “SIRT 2 inhibitor as anti-proliferation and apoptosis inducing agent in OCI-LAM53 cells” are not present in the manuscript. Please add. Furthermore why to perform these experiments the authors have chosen OCI-Ly8-LAM-53 cells. Please explain rationale